# Molecule Meets Protein Pocket
# 3D-Aware Molecular Optimization for Protein Targets

## Abstract

Lead optimization, refining drug candidates to improve binding to protein targets, is a key challenge in drug discovery. We introduce a novel 3D-aware generative architecture designed for fragment-level molecular optimization conditioned on the geometry of the protein's binding pocket. The model decomposes the molecule into a stable scaffold and generates new fragments using a Variational Autoencoder (VAE) and a SMILES-based transformer guided by local pocket structure. To handle the imbalance in fragment sizes, we incorporate a focal loss. On the CrossDock2020 benchmark, our proposed architecture outperforms prior approaches in generating diverse, novel, and chemically valid candidates with improved Vina scores while generalizing to unseen proteins.

## 1 Introduction

Molecular optimization, tuning a candidate molecule to improve binding to a target protein aims to modify the molecule while retaining its core structure. One frequent goal is to enhance binding to a specific protein target, but changes must also maintain favorable properties of the original compound. Recent machine learning approaches have shown promise in automating this process. Some treat it as a molecule-only task, ignoring the protein entirely and relying solely on SMILES strings or molecular graphs (Zhou et al., 2019; You et al., 2018; Barshatski & Radinsky, 2021). Some models incorporate protein sequences, such as FASTA (Kaminsky et al., 2023), but these lack the spatial detail necessary to capture molecular binding. Others leverage 3D protein information for docking and folding proteins (Varadi et al., 2022; Ketata et al., 2023), and some have explored graph-based neural networks that could embed 3D information from the ligand and the protein (Huang et al., 2024; Schneuing et al., 2024; Peng et al., 2022). However, these models are often diffusion-based, primarily designed for de novo generation as opposed to optimization, and require manual input, limiting automation and practical applicability.

To address these limitations, we introduce Molecular Docking-based Lead Optimization (MODOLO). Our primary contribution is methodological: we propose a specialized architecture tailored for 3D fragment generation conditioned on the geometry of the protein's binding pocket and the scaffold of the reference molecule. Given a target protein, a candidate molecule, and its docking pose within the binding site, our model represents this docked complex as a sparse graph. Nodes correspond to the molecule's atoms, the pocket's atoms, the alpha carbons of the protein's amino acid residues, and a special hole node representing the attachment point of the fragment. Edges connect nodes that lie within chemically meaningful distances, capturing relevant spatial interactions. A graph neural network encoder with grouped vector attention then learns a localized representation of this 3D chemical context. Next, we decompose the molecule into a stable scaffold using a hierarchical pruning strategy based on (Schuffenhauer et al., 2007), and creating "holes" where the remaining fragments were attached. For each hole, the pocket-aware embedding from the graph encoder is passed to a VAE-based transformer, which generates a 1D SMILES encoded replacement fragment predicted to bind more effectively within the pocket. Reattaching the generated fragments results in an optimized analogue that remains structurally close to the original molecule but is methodologically fine-tuned to the 3D shape of the target binding site.

To support generalization to unseen targets, we train MODOLO using a Vina score objective on the Cross-Dock2020 dataset (Francoeur et al., 2020). A focal loss ensures robust learning despite the imbalance in fragment distribution, while the VAE architecture allows for diverse fragment generation.

The experiments on synthetic CrossDocked dataset demonstrate that MODOLO can generate drug-like, synthesis-accessible, diverse molecules with high binding affinity against specific proteins and outperform the state-of-the-art (SOTA) models on multiple evaluation metrics.

The methodological contributions of this work are summarized as follows:

- We design a specialized generative architecture, MODOLO, that performs lead optimization by conditioning fragment generation on the geometric constraints of the protein binding pocket while preserving the core molecular scaffold.

- We introduce a novel conditioning scheme that utilizes a sparse heterogeneous graph representation of the protein-ligand complex, employing grouped vector attention to capture fine-grained spatial interactions.

- We demonstrate through extensive experiments on the CrossDock2020 benchmark that our proposed architecture achieves state-of-the-art performance, generating diverse, chemically valid, and high-affinity drug candidates that outperform existing diffusion and sequence-based baselines.

- We release our code and pre-trained models to support further research in structure-based drug design: `https://anonymous.4open.science/r/modolo-F13D`.

## 2 Related Work

In this work, we address the task of molecular optimization with the aim of generating chemically valid molecules that preserve the scaffold of an input molecule while improving its interaction with a specific protein. Unlike general molecular generation, which aims to design molecules with favorable properties from scratch, molecular optimization emphasizes structural similarity to a given molecule, creating a unique challenge. Our approach leverages 3D spatial information from the molecular docking of the molecule with the target protein to guide the optimization process.

**Protein-Free Optimization Methods**. Many molecular optimization models focus on modifying molecules without explicitly accounting for the protein target they are intended to bind. These approaches typically represent molecules using SMILES strings or molecular graphs (Blaschke et al., 2018; Gómez-Bombarelli et al., 2018; Harel & Radinsky, 2018; Olivecrona et al., 2017; Popova et al., 2018; You et al., 2018; Zhou et al., 2019; Fu et al., 2020; Jin et al., 2020; 2019; Liu et al., 2018; Seo et al., 2024; Shen et al., 2025). While such methods have been highly successful at generating syntactically valid and drug-like compounds, they do not directly incorporate information about the protein binding site, which can be important for optimizing target-specific bioactivity. To bridge the gap toward target-aware optimization, some approaches attempt to adapt to specific targets either by training a model strictly on target-specific data (Yang et al., 2021), or by iteratively utilizing external scoring oracles to guide the generation process (Reidenbach, 2024). However, a significant downside to these methods is that they are highly computationally expensive

Our method addresses this critical gap by explicitly modeling the 3D geometry of the protein binding pocket and conditioning the fragment generation process on spatial interactions between the molecule and the target. This enables the model to generate structurally compatible analogues that are more likely to bind effectively in real biological settings.

Related techniques such as scaffold hopping aim to discover structurally novel compounds by altering the core scaffold of known actives (Böhm et al., 2004; Zheng et al., 2021). While useful for scaffold diversity, such methods are less applicable to tasks, where preserving the original molecule's scaffold is often essential. In contrast, our approach explicitly retains the scaffold and focuses on generating new fragments tailored to the 3D shape of the protein binding site.

**Protein-Aware Generative Models.** Recent methods have begun incorporating protein structure into molecular generation, particularly in the context of de-novo drug design. Pocket2Mol (Peng et al., 2022) is one of the earlier works to directly condition molecule generation on 3D protein pocket structures. It employs a two-stage approach that first extracts pocket features using 3D convolutional networks and then generates molecules atom-by-atom or fragment by fragment, using an autoregressive model.

Other approaches leverage 3D equivariant diffusion or flow matching to generate entirely new compounds within the 3D pocket (Chen et al., 2025; Guan et al., 2023; Schneuing et al., 2025; Guan et al., 2024; Huang et al., 2024; Schneuing et al., 2024; Lin et al., 2025) typically initialize a prior distribution of atomic coordinates (or Gaussian noise) and iteratively denoise it into a final ligand. Additional works attempt to generate samples with higher affinity to the target via DPO loss (Gu et al., 2024; Cheng et al., 2024) and utilization of external oracles (Zhou et al., 2024). Within those works, some also tackle lead optimization, through partial diffusion (Schneuing et al., 2024) or by allowing the user to manually select a linker to replace of scaffold to preserve during the generative process Huang et al. (2024). However, partial diffusion usually discard the bonds of the original molecule, making it difficult to retain pharmacological properties and similarity to the original molecule, and human intervention limits automation.

Other approaches improve chemical realism by editing only small fragments of the molecule to preserve similarity. For instance, CFOM (Kaminsky et al., 2023) utilizes an encoding of the target protein sequence and outputs the fragments as SMILES strings, but consequently loses much of the physical context of binding. To better capture this context, other fragment-based methods incorporate explicit 3D structures. DeepFrag (Green et al., 2021) employs graph neural network to choose a fragment from an existing library, DiffDec (Xie et al., 2024) utilizes 3D graph diffusion specifically suited to add fragments to an existing scaffold, however, its iterative diffusion process remains computationally expensive during inference. MODOLO combines the structural rigor of geometric neural networks with the flexibility of Transformers in an efficient, one-shot architecture. It encodes the molecule-protein complex data within a 3D GNN and directly outputs SMILES fragments. By modeling the complex as a 3D graph and using GNNs with grouped vector attention, it effectively captures spatial interactions while ensuring chemical realism.

**Molecular Docking.** Molecular docking refers to the task of predicting how a small molecule fits into the 3D structure of a protein-essentially estimating the "pose" of the molecule within the binding site (Figure 2). There are two main approaches: classical search-based methods such as AutoDock Vina (Trott & Olson, 2010; McNutt et al., 2021), which use physics-inspired scoring functions and optimization algorithms to find likely poses, and newer machine learning-based methods that directly predict the docking pose using deep neural networks (Ketata et al., 2023; Ganea et al., 2021; Stärk et al., 2022). While docking methods are crucial for estimating how molecules interact with proteins, they are not designed to perform molecular optimization. Our work assumes a docking pose is already available and focuses instead on modifying the molecule itself to improve bioactivity-conditioned on the 3D geometry of the protein binding site. In other words, docking predicts how a molecule fits; we focus on how to optimize it for a better fit.

## 3 MODOLO Algorithm

Given a candidate molecule and a target protein, our goal is to optimize the molecule by modifying its peripheral fragments to improve binding affinity to the protein, while preserving its core structure. We assume as input a *docking pose* (see an example in Fig 2), a 3D assignment of atomic coordinates in $\mathbb{R}^3$ for both the protein and the molecule, representing how the molecule fits into the protein's binding pocket. From this complex, we construct a graph that captures spatial relationships between atoms in both the molecule and the protein. The molecule is then decomposed into a *scaffold* (the stable core) and one or more *fragments* (the modifiable parts). We formulate the optimization task as generating new fragments that, when reattached to the scaffold, yield an improved molecule that (1) fits the protein pocket more effectively and (2) remains similar to the original molecule.

MODOLO addresses this task through a modular pipeline (see Fig 1). First, with the 3D docking pose of the molecule in the protein's binding pocket, we build a sparse, heterogeneous graph representation of the molecule-protein complex, capturing geometric and biochemical features (see Section 3.1). Next, we extract chemically meaningful scaffolds of the molecule and identify the associated fragments to be regenerated (see

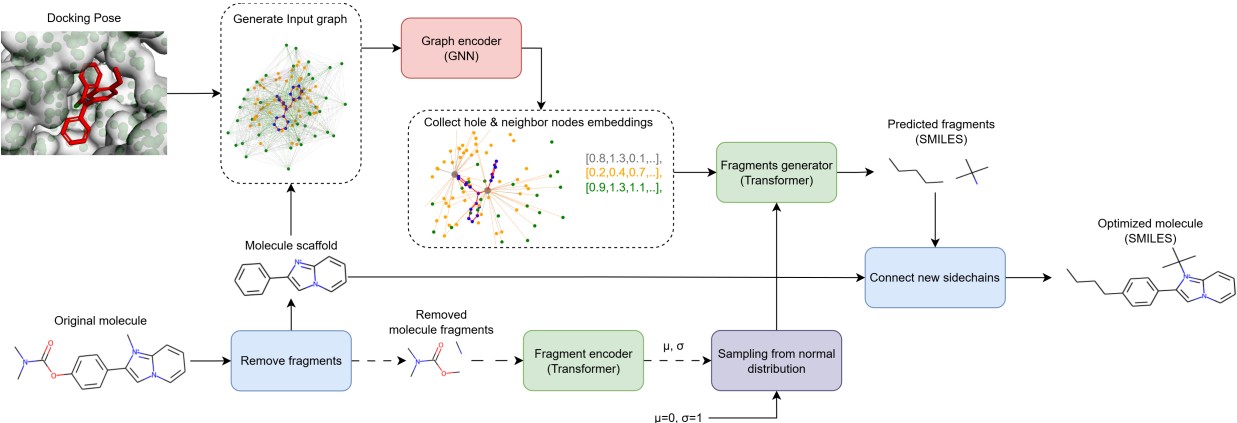

Figure 1: An overview of the Algorithm. The inputs a molecule and a docking pose of the molecule in a target protein. The molecule is split into a scaffold and fragments. The fragments are masked and the docking pose is fed into a graph based encoder. The encoded nodes of the graph are used as input to a text based fragment decoder that outputs the new, target specific fragments. Finally, the new fragments are reattached to the scaffold and the result is a new molecule that remains similar to its origin and has a stronger interaction with the target. The original, masked fragments are embedded and used as an input to a variation autoencoder. The dotted arrows represent flows of information that occur during training only.

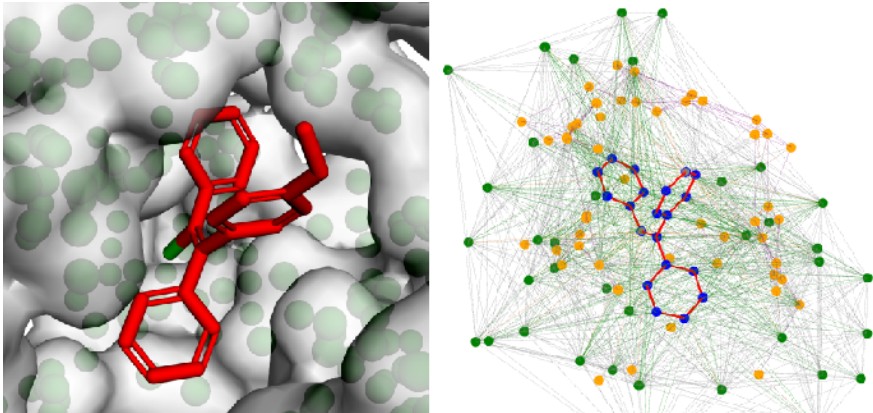

Figure 2: The image on the left depicts an input docking pose of a molecule within a pocket. The image on the right depicts the heterogeneous graph constructed from the docking pose. Ligand atoms are shown in blue, protein receptor atoms in orange, and amino acids in green. Hole nodes are shown in grey. The difference in connection distances (cutoffs) between receptor atoms and residues is illustrated by their respective edge lengths.

Section 3.2). These graphs are then processed using a geometric deep learning module with *grouped vector attention*, which learns spatial interactions across molecule and protein nodes (see Section 3.3). The original, masked fragments are embedded and used as an input to a variation autoencoder. Finally, a transformer decoder generates optimized fragments conditioned on the scaffold, the latent representation of the original fragments (during training only) and the protein context (see Section 3.4). In the following subsections, we detail each component of this pipeline.

## 3.1 Input Graph

To represent molecular docking structures, we construct heterogeneous geometric graphs (see an example in Fig 2) that encode ligand atoms, receptor amino acids (represented spatially by their central $\alpha$ carbon), receptor atoms, and a special **hole** node. The hole node is positioned at the atom connecting the fragment to the scaffold. To ensure computational efficiency and incorporate useful inductive biases, the graphs are sparsely connected based on distance-dependent cutoffs. Specifically, covalent and spatial interactions between ligand and receptor atoms are connected using a 5 Å cutoff, while receptor residues use larger cutoffs of 10 Å to capture long-range contextual information. Structure-based edges are defined by these proximity rules, while ligand bonds are preserved. Node features include biochemical properties, and ESM2 embeddings (Lin et al., 2023) are used for amino acid nodes. A detailed breakdown of the cutoff values and feature definitions is provided in appendix A.

## 3.2 Scaffold and Fragments Extraction

To systematically deconstruct each molecule into a scaffold and its fragments, we employ a scaffold-based decomposition strategy introduced by Kaminsky et al. (2023). The primary motivation is to preserve the core structural and functional features of the molecule while allowing controlled modification through fragment generation.

We begin by extracting the Murcko scaffold of the molecule using RDKit (Landrum, 2016). This scaffold captures the core ring systems and linkers essential to the molecule's chemical identity. However, relying solely on the Murcko scaffold may not always provide the optimal balance between preserving key properties and allowing sufficient flexibility for optimization. Therefore, we apply the hierarchical scaffold ordering rules proposed by (Schuffenhauer et al., 2007), implemented using ScaffoldGraph (Scott & Edith Chan, 2020), to rank and prune peripheral ring systems iteratively.

This pruning process removes less-characteristic substructures while considering topological and chemical characteristics such as ring size, heteroatom content, and attachment points. From the ranked list of scaffold candidates, we select the smallest scaffold whose molecular weight constitutes at least 70% of the original compound. The 70% threshold was empirically chosen as it ensures that the retained scaffold captures the majority of the molecule's pharmacophoric features, while still allowing meaningful optimization through fragment addition.

The removed substructures are defined as fragments. Formally, for a molecule $m$, we denote its set of fragments as $F = \{f_1, \ldots, f_n\}$, where each $f_i$ is a disconnected component removed during scaffold extraction. For each fragment $f_i$, a hole $h_i$ is defined as an atom in the scaffold that has a bond with it in the original molecule. Each molecule has a single scaffold but may yield multiple fragments, each fragment has a single hole, due to the extraction process not splitting rings.

An illustrative example of the scaffold extraction process, including weight ratios and intermediate scaffolds, is shown in Figure 3.

## 3.3 Interaction Layers

After constructing the heterogeneous graph representing the molecule protein complex and decomposing the molecule into scaffold and fragments, the next step is to model the spatial and chemical interactions within this graph. To effectively capture how atoms and residues interact in 3D space, the model must go beyond simple message passing and incorporate directional, context-dependent relationships between nodes.

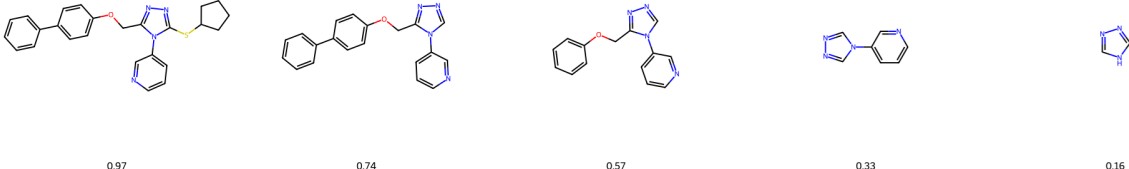

0.97          0.74          0.57          0.33          0.16

Figure 3: An example of the molecule scaffold extraction process in steps. The original molecule is shown on the left, every following scaffold is the result of a single fragment removal in the iterative process. Below are the ratios between the molecular weight of the scaffold and the original molecule.

Our model utilizes vector attention, a mechanism where attention weights are vectors that modulate individual feature channels of node embeddings (Zhao et al., 2021). This approach allows the network to learn fine-grained interactions between nodes in the input graph, which is crucial for capturing the complex spatial and chemical relationships present in molecular docking structures.

The interaction layer operates as follows: For each node $i$ in the graph, its updated embedding $x_i$ is computed by aggregating information from its neighbors $x_j \in X(i)$, where $X(i)$ is the set of neighboring nodes and $E(i)$ is the set of edges connecting $i$ to its neighbors. The relation between node $i$ and each neighbor $j$ is modeled using a subtraction-based relation function, which enhances the model's ability to capture directional and contextual differences.

The attention weight $w^{ij}$ for each neighbor is computed as:

$$w^{ij} = \gamma \left( \varphi(x_i) - \psi(x_i, e_k, x_j) \right)$$

where $\varphi$ and $\psi$ are multi-layer perceptrons (MLPs) that encode the node and edge features, $e_k$ is the edge feature between $i$ and $j$, and $\gamma$ is an activation function (e.g., sigmoid or softmax) that normalizes the attention weights.

The updated node embedding is then given by:

$$x_i = \sum_{\substack{x_j \in X(i) \\ e_k \in E(i)}} w^{ij} \cdot \alpha(x_i, e_k, x_j)$$

where $\alpha$ is a linear transformation applied to the concatenated features of $x_i$, $e_k$, and $x_j$.

While vector attention provides expressive power, it introduces a large number of parameters, especially in deep networks. To address this, we employ **grouped vector attention** (Wu et al., 2022), which divides the feature channels into groups. Each group shares a single attention weight, reducing the number of parameters and improving generalization.

Formally, for $n$ feature channels divided into $g$ groups, the updated embedding for the $m$-th channel is:

$$x_{i_m} = \sum_{\substack{x_j \in X(i) \\ e_k \in E(i)}} w^{ij}_{\lfloor m/g \rfloor} \cdot \alpha(x_i, e_k, x_j)_m$$

where $w^{ij} \in \mathbb{R}^{n/g}$ is the vector of attention weights for each group, and $\alpha(x_i, e_k, x_j)_m$ denotes the $m$-th channel of the transformed features.

This grouped attention mechanism enables the model to efficiently capture complex interactions in large graphs while maintaining computational tractability and strong generalization performance.

### 3.4 Fragments Generator

Once the model has encoded the spatial and chemical context of the scaffold and surrounding protein environment through the interaction layers, the next step is to generate optimized molecular fragments.

The Fragments Generator architecture consists of a variational Autoencoder (VAE) (Kingma & Welling, 2019) that conditions the generation on the structural context. The encoder of the VAE takes the fragment SMILES (during training) and embeds them using a transformer encoder. The decoder is a transformer decoder (Vaswani et al., 2017). For input graph $\mathcal{G}$, hole $h$ and fragment $f$, the input to the decoder includes:

- The embeddings of nodes that were connected to hole $h$ in graph $\mathcal{G}$, which are added as the *memory* of the decoder.

- hole $h$ embedding is introduced as the *first target token* for the generation sequence.

The decoder outputs the new fragment as a SMILES string, maximizing the likelihood of the valid fragment given the context. Note that each SMILES fragment is generated separately.

The goal of the model is to regenerate the fragment $f$ SMILES from the masked ligand in a protein-ligand complex. To this end, our loss function is a combination of a reconstruction loss (Section 3.4.1) and a VAE loss (Section 3.4.2):

$$L_{total}(f, \mathcal{G}, h) = L_{focal\_recon}(f, \mathcal{G}, h) + \beta \cdot L_{KL}(f)$$

where $\beta$ is a hyperparameter that controls the trade-off between the two terms.

#### 3.4.1 Reconstruction Loss

The reconstruction loss is defined as the negative log-likelihood of the true fragment SMILES given the scaffold and hole. The reconstruction loss of a single fragment token is given by:

$$L_{recon}(f, \mathcal{G}, h, t) = -\log p(f_t | f_{<t}, h, \mathcal{G})$$

where $f_t$ is the $t$-th token in the true fragment $f$ SMILES, $f_{<t}$ are the previous tokens of the SMILES. Due to the data imbalance, we employ a **Focal Loss** to handle the data imbalance, particularly the prevalence of single-atom fragments. The focal loss is defined as:

$$L_{focal\_recon}(f, \mathcal{G}, h, t) = (1 - p(f_t | f_{<t}, h, \mathcal{G}))^\gamma \cdot L_{recon}(f, \mathcal{G}, h, t)$$

where $\gamma$ is the focusing parameter. The focal loss puts more emphasis on hard-to-classify examples, where the model is less confident in its predictions, which are the non-single-atom fragments in this case. To further battle the imbalance, the total loss of a fragment is computed as the mean of the focal loss of each token, and weighted by the length of the fragment:

$$L_{focal\_recon}(f, \mathcal{G}, h) = \frac{\log T}{T} \sum_{t=1}^{T} L_{focal\_recon}(f, \mathcal{G}, h, t)$$

where $T$ is the length of the fragment $f$.

#### 3.4.2 KL divergence

The second component is the Kullback-Leibler (KL) divergence, which regularizes the learned latent distribution to approximate the standard normal prior $p(z) = \mathcal{N}(0, I)$:

$$L_{KL}(f) = D_{KL}\left(q_\phi(z|f) \middle\| p(z)\right) = -\frac{1}{2} \sum_{i=1}^{D} \left(1 + \log \sigma_i^2 - \mu_i^2 - \sigma_i^2\right)$$

This constraint ensures a continuous latent space for valid sampling. However, to prevent *posterior collapse*, where the autoregressive decoder ignores the latent code $z$, we employ **Cyclical KL Annealing** (Fu et al., 2019). By cyclically increasing the KL weight $\beta$ from 0 to 1, we allow the decoder to learn reconstruction using unconstrained latent information before gradually enforcing the regularization.

### 3.5 Inference

During inference, we sample from the prior latent distribution and decode the SMILES string conditioned on the hole and context embeddings. The validity of the generated token is checked, in case of an error, the fragment is resampled.

# 4 Experimental Setup

### 4.1 Datasets

We train and evaluate our model using the standard split of the CrossDocked2020 dataset (Francoeur et al., 2020). Originating from the Protein Data Bank, this dataset initially comprises 22.5 million docked protein-ligand pairs, generated using smina via the Pocketome workflow.

To ensure data quality and consistency with prior work (Huang et al., 2024; Xie et al., 2024), we utilize a filtered subset restricted to high-quality binding poses with a root-mean-squared deviation (RMSD) of less than 1Å. To prevent data leakage and ensure structural diversity, the dataset was clustered at 30% sequence identity using MMseqs2, as proposed by Luo et al. (2021).

### 4.2 Model Implementation

Our model is implemented using PyTorch and PyTorch Lightning. The 3D pocket encoder is a 4-layer graph neural network (GNN), where each layer includes grouped vector attention with 8 groups and applies dropout with a rate of 0.3. The decoder and encoder are both 2-layer Transformer with 4 attention heads per layer. Optimization is performed using the Adam optimizer(Kingma & Ba, 2015) with an initial learning rate of $10^{-4}$ and a weight decay of $10^{-4}$. The loss hyper parameters are $\beta = 0.1, \gamma = 2, M = 4$.

Experiments were run on a server equipped with 2 NVIDIA L40 GPUs (each with 48 GB of VRAM). Training on the CrossDock2020 dataset required approximately 6 hours to complete, including both training and validation phases. Inference for molecule generation takes 0.3 seconds per molecule on average using a single GPU.

### 4.3 Baselines

1. **DiffSBDD** (Schneuing et al., 2024). A diffusion-based model that generates molecules based on the 3D structure of the target pocket. The model can generate molecules via a denoising process, and also can optimize molecules by partially noising and denoising the reference molecule. We use the optimization method.

2. **CFOM** (Kaminsky et al., 2023). Lead optimization transformer encoder decoder model that extracts and encodes scaffold molecule SMILES and protein FASTA. Outputs SMILES of interaction optimized side-chains for the molecule.

3. **Pocket2Mol** (Peng et al., 2022). A GNN-based method that generates 3D molecules atom-wise under the context of the pocket. The method is designed for molecule generation, and was adjusted for scaffold conditioned fragment generation. The setup is shown in B.

4. **DiffDec** (Xie et al., 2024). A diffusion based model that generates a fragment based on the 3D structure of the target pocket and an extracted a scaffold that is acquired by slicing the reference molecule with a set of reaction based rules.

5. **PMDM** (Huang et al., 2024). A diffusion model capable of generating molecules while preserving a manually defined core scaffold. To ensure a fair comparison, we provide this model with the exact same scaffold extracted by our hierarchical pruning algorithm.

6. **Random Baseline**. A naive baseline that randomly samples replacement sidechains from the empirical distribution of fragments found in our training set. This serves to establish a lower bound for optimization performance and validate the learned spatial conditioning of our model.

### 4.4 Ablations

To better understand the contribution of each component in our model, we conducted a comprehensive ablation study. For each ablation, we retrained the model under identical conditions and evaluated its performance on the CrossDocked benchmark using the same metrics as in the main experiments. Specifically, we evaluated the following variants:

- **No alpha carbons**: In this variant, we excluded the $\alpha$-carbon nodes representing amino acid residues from the input graph, reducing the amount of structural and contextual information about the protein pocket. As a result, the model relies solely on the atomic-level representation of the protein, potentially limiting its ability to capture higher-level spatial relationships and long-range interactions that are important for accurate molecular optimization. Compared with the graph in Figure 2, the graph without alpha carbons is missing the green nodes.

- **No VAE**: We performed an ablation study where the VAE module was removed from the architecture. In the ablated model, the 3D context from the GNN is fed directly into the transformer encoder. This forces the decoder to map the 3D structural context of the hole and its neighbors directly to a single SMILES sequence, effectively removing the model's ability to sample diverse chemical outputs for a given geometric configuration.

- **No attention groups**: In this variant, we eliminated the use of grouped vector attention in the interaction layers, reverting to standard vector attention. This increases the number of parameters in the model and may lead to overfitting, especially in data-scarce scenarios. Grouped attention is designed to improve generalization by sharing attention weights across groups of channels, so its removal tests the importance of this regularization mechanism.

- **No ESM2 protein embeddings**: We removed the high dimensionality embeddings for the receptor nodes in the input graph. The purpose of those embeddings was to add global context of the protein to the GNN, because during the graph encoding process, only the local environment close to the target pocket was given to the model. The ablation will evaluate the contribution of those embeddings to the final result.

- **Conditional Prior**: To test the effects of the prior, we sample the latent variable from $p(z|ref)$ instead of $p(z)$, where $ref$ is an encoded SMILES representation of the reference molecule. In this case, the KL divergence loss component is $D_{KL}\Big(q_\phi(z|f, ref)\Big|\Big|p(z|ref)\Big)$

- **Noisy docking Pose:** In this ablation, we introduce controlled perturbations to the molecular docking pose by applying random rotational transformations and random translations of different magnitudes to the molecular graph. This experiment is designed to evaluate the robustness of MODOLO to inaccuracies in the prediction of the docking pose.

### 4.5 Evaluation Metrics

To evaluate the structural fit within the protein pocket, we utilize the Vina Score, which estimates the binding affinity between the ligand and the target. The score is calculated by redocking the generated molecule to the protein pocket using GNINA (McNutt et al., 2021), moreover, the pose score given to the generated pose, that evaluates its quality is also shown. Regarding drug-likeness and developability, we report QED (Quantitative Estimation of Drug-likeness), SA (Synthetic Accessibility), and LogP (octanol-water partition coefficient), considering the range of -0.4 to 5.6 as optimal.

We measure the similarity between the generated molecules and the original molecule, with the Tanimoto similarity (Bajusz et al., 2015) between the two Morgan fingerprints of the molecules.

We also measure success, defined by:

$$(Sim(org, opt) > 0.4) \wedge (Vina(org) > Vina(opt))$$

This definition of success combines constraints from prior works, specifically drawing the binding affinity improvement from one approach (Huang et al., 2024) and the similarity threshold from others (Jin et al., 2019; Kaminsky et al., 2023; Fu et al., 2021). We adopt the 0.4 Tanimoto similarity threshold to maintain consistency and ensure a fair comparison with these prior models. Furthermore, this value represents a well established empirical boundary, enforcing a similarity above 0.4 ensures that the optimized molecule remains structurally related to the original lead, representing a valid derivative or scaffold hop (Steshin, 2023; Franco et al., 2014). To provide a comprehensive evaluation, we report additional optimization results across a range of different similarity thresholds in Appendix D.

For each molecule in the test set, 20 new molecule where generated. The measured metric are averaged over all generated molecules.

## 5 Empirical Results

### 5.1 Baselines

The results of our evaluation on the test set are reported in Table 1. We observed that MODOLO got the best results in in affinity and success rate. According to the Vina score, MODOLO handles the task of keeping a high binding affinity to the target pocket while preserving a high structural similarity to the original molecule.

DiffSBDD achieved the lowest success rates among the compared methods. This is primarily due to its diffusion-based approach, which generates molecules by iteratively perturbing atomic features and 3D coordinates. During this process, explicit chemical bonds are not preserved. Instead, the final molecular graph is reconstructed by inferring bonds based on interatomic distances and chemical heuristics, such as covalent radii and valence rules. As a result, the generated molecules often lack sufficient structural similarity to the original compounds, leading to lower optimization success. In contrast, DiffDec was designed to keep intact the exact components of the scaffold and only noise and denoise the fragments around it. As can be seen in the Results, DiffDec accumulated results that are comparable to MODOLO. While achieving comparable performance, our approach inherently avoids the iterative sampling steps characteristic of diffusion models thus resulting in computational efficiency during inference.

CFOM was explicitly trained for the molecular optimization task with the objective of enhancing molecular activity against the target protein, it has a very high similarity, which could be due the lack of a VAE component within the architecture.

The logP value of MODOLO falls within the compliance range ($-0.4 < LogP < 5.6$), which implies that the molecules generated by MODOLO hold greater promise as drug candidates, which is crucial for clinical trials.

Fig. 4 presents a representative case study of a molecule optimized by MODOLO. As shown, the generated analogue strictly preserves the original scaffold pose while evolving the fragment to better occupy the binding cavity, resulting in improved geometric complementarity and a lower Vina score compared to the starting compound.

### 5.2 Ablations

The results of the ablation study are reported in Table 2. As can be seen, the full model achieved the highest and Vina score, which are the main metrics of interest. This confirms the effectiveness of combining all architectural and training components.

The removal of the grouped vector attention led to a higher model complexity, which resulted in a longer training time, but did not improve the performance. The degradation in performance observed in the VAE ablation highlights the inherent multi modality of the lead optimization task. The relationship between a specific 3D protein-ligand interface and a valid chemical fragment is one-to-many, multiple distinct chemical groups can often satisfy the same geometric and physical constraints. By removing the latent vector $z$, the model loses the capacity to represent this distribution of possibilities. Instead of learning a manifold of valid

Table 1: metrics of MODOLO and baseline models on the CrossDocked dataset.

| | Affinity ↓ | pose Score ↑ | Similarity ↑ | SA ↓ | LogP | QED ↑ | Success Rate ↑ |
|---|---|---|---|---|---|---|---|
| Ref | -7.50 ±24.04 | 0.83 ±0.02 | - | 3.45 ±1.89 | 2.85 ±6.05 | 0.50 ±0.05 | - |
| MODOLO | **-7.56** ±11.45 | 0.74 ±0.03 | 0.53 ±0.02 | 3.49 ±2.31 | 3.00 ±5.75 | 0.52 ±0.05 | **0.38** ±0.24 |
| CFOM | -7.19 ±30.19 | 0.74 ±0.03 | 0.54 ±0.04 | **3.11** ±1.56 | 3.63 ±2.84 | 0.56 ±0.03 | 0.27 ±0.20 |
| DiffSBDD | -5.78 ±78.60 | 0.63 ±0.04 | 0.15 ±0.00 | 4.29 ±1.42 | 3.31 ±3.37 | 0.59 ±0.04 | 0.01 ±0.01 |
| DiffDEC | -7.12 ±20.94 | **0.79** ±0.03 | **0.57** ±0.02 | 3.45 ±1.76 | 2.49 ±2.86 | **0.63** ±0.05 | 0.27 ±0.20 |
| Pocket2Mol | -6.98 ±3.49 | 0.64 ±0.03 | 0.09 ±0.01 | 4.53 ±1.74 | 1.66 ±4.38 | 0.56 ±0.02 | 0.00 ±0.00 |
| PMDM | -7.04 ±35.54 | 0.65 ±0.03 | 0.21 ±0.01 | 4.42 ±2.17 | 3.62 ±3.88 | 0.54 ±0.04 | 0.01 ±0.01 |
| Frags | -6.96 ±31.23 | 0.63 ±0.05 | 0.44 ±0.03 | 3.62 ±2.78 | 2.73 ±4.37 | 0.47 ±0.05 | 0.24 ±0.18 |

Best and second-best results are highlighted in bold and underlined, respectively. Asterisks (*) in the Affinity column denote results that are not statistically significantly different from the best performance (t-test, $p > 0.05$).

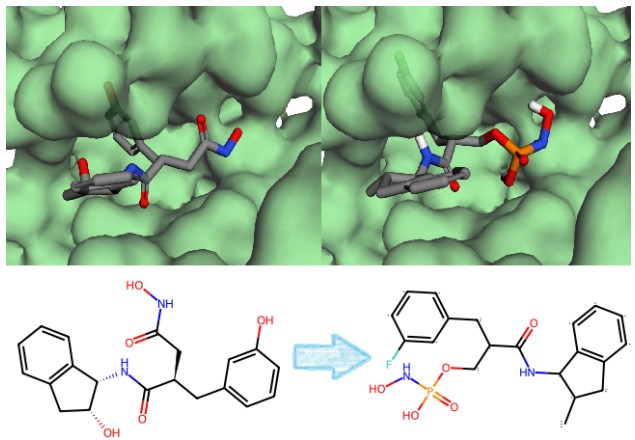

Figure 4: On the left, the original molecule and its binding pose (with a score of -9.4 kcal/mol). On the right, the generated molecule and its binding pose (with a score of -9.9 kcal/mol). The target protein is Q9UNA0 (Human).

fragments, the deterministic model is forced to approximate a single output for given geometric inputs, likely resulting in "averaging" effects where the generated fragments fail to capture the necessary chemical diversity or validity inherent in the training data. The most apparent degradation in performance was observed during the masking of the $\alpha$-carbon atoms, where the loss of the protein residue structure severely impaired the model's ability to interpret the global topology of the binding pocket. Consequently, this led to a decrease in Vina scores, indicating that the coarse-grained spatial constraints provided by the $\alpha$-carbons are essential for guiding the generation of high-affinity ligands. The removal of ESM2 embeddings from the protein's $\alpha$ carbon nodes resulted in a slightly worse affinity and success rate. The Conditional prior ablation showed no significant difference in terms of affinity, but had a higher similarity, due to the input molecule that effected the prior from which the fragments where sampled.

The introduction of docking pose noise revealed a disconnect between predicted binding affinity and structural quality in Table 3. While the Vina scores remained remarkably stable across all noise levels, suggesting the model can still locate energetic minima even when displaced, the physicochemical profile of the generated molecules degraded. We observed a notable drift from the reference active structure, as noise increased, the generated fragments became less similar to the masked, ground truth fragments, indicating that the model lost the specific geometric guidance necessary to reproduce them. This structural deviation was accompanied by a slight deterioration in both synthetic accessibility (SA) and QED. Most significantly, we observed a steady decline in lipophilicity, with LogP dropping. This downward trend pushes some of the generated compounds out of the optimal lipophilicity range for drugs $1 < LogP < 3$ (Waring, 2010; Landry & Crawford, 2019), suggesting that as the generation centroid shifts from the hydrophobic pocket into the

Table 2: metrics of MODOLO and ablations on the CrossDocked dataset.

| | Affinity ↓ | pose Score ↑ | Similarity ↑ | SA ↓ | LogP | QED ↑ | Success Rate ↑ |
|---|---|---|---|---|---|---|---|
| Ref | -7.43 ±11.64 | 0.81 ±0.03 | - | 3.65 ±1.63 | 0.97 ±8.03 | 0.48 ±0.05 | - |
| MODOLO | **-7.09** ±7.32 | 0.73 ±0.03 | 0.53 ±0.04 | 3.60 ±1.98 | 1.46 ±7.99 | 0.51 ±0.05 | **0.30** ±0.21 |
| No groups | -6.64 ±11.30 | 0.74 ±0.03 | 0.46 ±0.03 | 3.50 ±1.95 | 0.67 ±7.69 | 0.55 ±0.04 | 0.21 ±0.17 |
| No VAE | -6.44 ±10.69 | 0.73 ±0.02 | 0.45 ±0.03 | 3.43 ±2.04 | 1.36 ±6.41 | 0.54 ±0.02 | 0.20 ±0.16 |
| No $\alpha$-carbons | -6.41 ±7.83 | 0.73 ±0.03 | 0.49 ±0.03 | **3.35** ±1.89 | 1.30 ±8.07 | **0.56** ±0.03 | 0.20 ±0.16 |
| No ESM | -6.81 ±10.88 | 0.74 ±0.03 | 0.55 ±0.03 | 3.52 ±1.83 | 1.47 ±6.89 | 0.52 ±0.05 | 0.27 ±0.20 |
| Cond prior | *-7.08 ±8.62 | **0.75** ±0.03 | **0.57** ±0.04 | 3.63 ±1.91 | 1.00 ±8.10 | 0.50 ±0.05 | 0.28 ±0.20 |

Best and second-best results are highlighted in bold and underlined, respectively. Asterisks (*) in the Affinity column denote results that are not statistically significantly different from the best performance (t-test, $p > 0.05$).

Table 3: performance metrics of MODOLO evaluated under varying levels of docking pose noise.

| Metric | Ref | Origin Pose | 4Å | 8Å |
|---|---|---|---|---|
| Affinity ↓ | -7.42 ±12.04 | **-7.05** ±7.46 | -6.93 ±10.93 | *-7.04 ±9.03 |
| pose Score ↑ | 0.81 ±0.03 | **0.73** ±0.04 | 0.69 ±0.04 | 0.69 ±0.04 |
| Similarity ↑ | - | **0.54** ±0.04 | 0.52 ±0.04 | 0.51 ±0.03 |
| SA ↓ | 3.61 ±1.68 | **3.60** ±2.01 | 3.75 ±2.00 | 3.69 ±2.05 |
| LogP | 1.10 ±8.30 | 1.51 ±8.11 | 1.19 ±8.45 | 1.13 ±7.99 |
| QED ↑ | 0.48 ±0.05 | **0.52** ±0.05 | 0.48 ±0.05 | 0.49 ±0.05 |
| Success Rate ↑ | - | **0.30** ±0.21 | 0.28 ±0.20 | 0.27 ±0.20 |

The Origin Pose column represents the original docking pose. The 4Å and 8Å columns represent docking poses perturbed by random translations (4 Å and 8 Å, respectively) and random rotations. Asterisks (*) in the Affinity column denote results that are not statistically significantly different from the best performance (t-test, $p > 0.05$).

solvent, the model compensates by producing overly polar structures that fall outside the optimal profile for our target.

## 6 Conclusion

In this work, we introduced MODOLO, a novel architecture for molecular lead optimization that leverages both the 3D structure of protein binding pockets and the SMILES representation of molecules. By decomposing molecules into scaffold and fragments, and utilizing a graph-based encoder alongside a transformer decoder, our approach enables the generation of structurally similar molecules with enhanced protein interactions. The integration of grouped vector attention, the heterogeneous graph structure, and the variational autoencoder enabled the model to learn a more compact and disentangled representation of the molecules, which improved the model's ability to generate structurally similar molecules with enhanced protein interactions. Our experiments on the CrossDock2020 benchmark demonstrated that MODOLO consistently outperforms state-of-the-art baselines. Looking forward, future research could explore extending the architecture to multi-objective optimization, incorporating additional chemical properties such as toxicity or solubility, and adapting the model for scaffold hopping or de novo drug design. Further investigation into integrating active learning strategies to maximize performance with minimal labeled data is also promising.

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

## A  Construction of Input Graph

Molecular docking structures are represented as heterogeneous geometric graphs with nodes representing ligand (heavy) atoms, receptor residues (located in the position of the $\alpha$-carbon atom), and receptor (heavy) atoms. To build the radius graph, we connect nodes using cutoffs that are dependent on the types of nodes they are connecting:

1. Ligand atoms-ligand atoms, receptor atoms-receptor atoms, and ligand atoms-receptor atoms interactions all use a cutoff of 5 Å, standard practice for atomic interactions. For the ligand atoms-ligand atoms interactions we also preserve the covalent bonds as separate edges with some initial embedding representing the bond type (single, double, triple and aromatic). For receptor atoms-receptor atoms interactions, we limit at 8 the maximum number of neighbors of each atom.

2. Receptor residues-receptor residues use a cutoff of 15 Å with 24 as the maximum number of neighbors for each residue.

3. Receptor residues-ligand atoms use a cutoff of 10 Å.

4. Receptor residues are connected to the receptor atoms that form the corresponding amino-acid.

Receptor residue features include the amino acid identity and a language model embedding derived from ESM2 (Lin et al., 2023). Ligand atom features include atomic number, chirality, degree, formal charge, implicit valence, number of attached hydrogens, number of radical electrons, hybridization state, aromaticity, number of rings, and six binary indicators for membership in rings of size 3 through 8. The edges are encoded based on the distance between the nodes, and the type of the bond (in ligand-ligand edges).

Moreover, during generation, for each masked fragment, a "hole" is created in the graph at the position of the scaffold's atom, to which the fragment is connected. The node is connected to all the neighbors of atoms in the masked fragment

## B  Adjustment of Pocket2Mol

The original Pocket2Mol model was designed for de novo 3D structure-based drug design, autoregressively generating molecules from scratch starting from a single sampled atom within an empty protein pocket. However, because our work focuses on lead optimization by expanding an existing core, a direct comparison with the unmodified de novo model would be misaligned.

To ensure a fair and direct evaluation, we adapted Pocket2Mol's initialization phase to support scaffold extension. Instead of beginning with an empty pocket, we seed the generation process with the 3D coordinates and atomic features of the extracted input scaffold, which was extracted with the method detailed in 3.4. Pocket2Mol then bypasses its initial sampling steps and directly expands upon this provided sub-graph, iteratively generating new atoms and bonds outward from the given scaffold.

Table 4: metrics of MODOLO and baseline models on the PoseBuster subset.

| | Affinity ↓ | pose Score ↑ | Similarity ↑ | SA ↓ | LogP | QED ↑ | Success Rate ↑ |
|---|---|---|---|---|---|---|---|
| Ref | -9.69 ±4.64 | 0.91 ±0.02 | - | 3.19 ±0.65 | 2.41 ±7.07 | 0.53 ±0.04 | - |
| MODOLO | **-9.34** ±3.67 | 0.84 ±0.04 | 0.56 ±0.02 | **3.28** ±0.75 | 1.93 ±5.69 | 0.52 ±0.05 | **0.25** ±0.19 |
| CFOM | -8.20 ±41.08 | 0.79 ±0.03 | 0.50 ±0.02 | 3.43 ±0.61 | 2.12 ±3.73 | 0.51 ±0.04 | 0.18 ±0.15 |
| DiffDEC | -8.55 ±11.08 | **0.86** ±0.02 | **0.57** ±0.04 | 3.36 ±0.57 | 1.95 ±5.01 | 0.61 ±0.05 | 0.18 ±0.15 |
| DiffSBDD | -7.35 ±39.06 | 0.64 ±0.05 | 0.15 ±0.01 | 4.06 ±0.93 | 3.08 ±2.92 | **0.62** ±0.04 | 0.00 ±0.00 |
| Pocket2Mol | -6.01 ±58.83 | 0.63 ±0.02 | 0.15 ±0.01 | 4.90 ±0.46 | -0.96 ±5.16 | 0.46 ±0.03 | 0.03 ±0.03 |
| PMDM | -5.58 ±148.72 | 0.62 ±0.05 | 0.19 ±0.01 | 4.27 ±0.99 | 2.04 ±3.33 | 0.57 ±0.04 | 0.04 ±0.04 |
| Frags | -6.86 ±85.77 | 0.68 ±0.06 | 0.45 ±0.02 | 3.54 ±0.96 | 2.05 ±3.22 | 0.48 ±0.06 | 0.16 ±0.14 |

Best and second-best results are highlighted in bold and underlined, respectively. Asterisks (*) in the Affinity column denote results that are not statistically significantly different from the best performance (t-test, $p > 0.05$).

## C Inference on additional dataset

While our primary models were trained and evaluated on the CrossDocked2020 dataset, we conducted an additional evaluation on a diverse 50-complex subset of the PoseBusters benchmark (Buttenschoen et al., 2024) to rigorously assess out-of-distribution generalizability. Because the PoseBusters dataset is composed exclusively of high-quality experimental crystal structures published since 2021, it provides a strict temporal separation from the CrossDocked2020 training distribution. Evaluating on this temporally isolated and completely independent benchmark ensures that the model's ability to capture and optimize 3D protein-ligand interactions is not overfitted to the structural distribution of historical training data, demonstrating true generalization to novel, unseen targets. The results of the evaluation are presented in Table 4. Similarily to the results in the main results Table 1, MODOLO achieves the best affinity and success rate.

## D Ablation on similarity thresholds and additional scaffolds

To provide a more comprehensive evaluation of optimization success, we extend the analysis in two directions. First, we broaden the similarity-based evaluation by considering additional metrics and varying the success thresholds defined over them, beyond the main setting reported in the primary tables. Table 5 summarizes the performance of all baselines under different Morgan fingerprint-based Tanimoto similarity thresholds. We also report results based on ChemNet similarity across multiple thresholds in Table 6. ChemNet yields a continuous representation trained to predict chemical and biological properties, providing a complementary perspective on structural and functional relatedness. Second, we ablate the effect of replacing the default scaffold-conditioning method with the alternative method proposed by Xie et al. (2024), which we denote as $MOLODO_D$.

As expected, increasing the strictness of the similarity threshold reduces the overall success rate across all models; generating candidates with improved binding affinity while maintaining near-identical topologies is inherently more difficult. Notably, MODOLO achieves the highest success rates across the majority of thresholds in both the structural Tanimoto (Table 5) and functional ChemNet (Table 6) evaluations.

An interesting pattern emerges at the highest Tanimoto similarity thresholds, where $MOLODO_D$, i.e., MOLODO using DiffDec-generated base scaffolds, achieves the best performance. This behavior is likely driven by the stricter fragmentation rules of DiffDec. Unlike MOLODO and CFOM, which permit an unbounded number of cuts, DiffDec limits the number of cuts in the original molecule to four. In addition, it constrains properties of the removed fragments, including molecular weight, heavy atom count, rotatable bonds, hydrogen bond acceptors, and hydrogen bond donors. These restrictions force the retained scaffold to remain highly similar to the original lead, resulting in stronger similarity preservation. This is particularly advantageous under higher similarity thresholds, where success requires the optimized molecule to remain closer to the original lead. In contrast, MOLODO's unconstrained fragmentation enables more substantial scaffold hopping and broader exploration of chemical space. This flexibility is important for escaping

Table 5: Success Rates of MODOLO and baseline models on the CrossDocked dataset with varying thresholds on Tanimoto similarity.

|  | 0.2 | 0.4 | 0.6 | 0.8 |
|---|---|---|---|---|
| MODOLO | **0.44** | **0.38** | 0.14 | 0.02 |
| $MODOLO_D$ | 0.28 | 0.27 | **0.22** | **0.08** |
| CFOM | 0.32 | 0.28 | 0.20 | 0.02 |
| DiffSBDD | 0.05 | 0.01 | 0.00 | 0.00 |
| DiffDEC | 0.30 | 0.29 | 0.19 | 0.04 |
| Pocket2Mol | 0.09 | 0.00 | 0.00 | 0.00 |
| PMDM | 0.19 | 0.01 | 0.00 | 0.00 |
| Frags | 0.34 | 0.24 | 0.09 | 0.00 |

Best and second-best results are highlighted in bold and underlined, respectively.

Table 6: Success Rates of MODOLO and baseline models on the CrossDocked dataset with varying thresholds on ChemNet similarity.

|  | 0.2 | 0.4 | 0.6 | 0.8 |
|---|---|---|---|---|
| MODOLO | **0.45** | **0.45** | **0.38** | **0.19** |
| $MODOLO_D$ | 0.28 | 0.28 | 0.25 | 0.16 |
| CFOM | 0.33 | 0.33 | 0.30 | 0.16 |
| DiffSBDD | 0.25 | 0.12 | 0.02 | 0.00 |
| DiffDEC | 0.30 | 0.30 | 0.24 | 0.11 |
| Pocket2Mol | 0.15 | 0.12 | 0.04 | 0.02 |
| PMDM | 0.39 | 0.32 | 0.16 | 0.02 |
| Frags | 0.37 | 0.33 | 0.20 | 0.04 |

Best and second-best results are highlighted in bold and underlined, respectively.

local optima in docking-based optimization, and likely explains its stronger performance at lower similarity thresholds, where greater structural deviation is permitted.

The advantage of $MOLODO_D$ at high Tanimoto thresholds does not transfer as clearly to ChemNet-based evaluation, likely because ChemNet reflects higher-level functional and pharmacological similarity rather than strict fingerprint overlap. Consequently, preserving a scaffold closer to the original lead is especially rewarded by Tanimoto similarity, but not necessarily by ChemNet similarity, where molecules with greater structural deviation may still remain close in embedding space.

