# OpenReview forum: "Molecule Meets Protein Pocket 3D-Aware Molecular Optimization for Protein Targets"
_TMLR — Rejected by TMLR_

### Review · Reviewer_RN2K · 2026-03-16

**Summary Of Contributions:**

The authors introduce MODOLO, a 3D-aware generative framework designed for molecular lead optimization. Instead of generating entirely new molecules from scratch, the model retains the core scaffold of an input molecule and generates new peripheral fragments conditioned on the 3D geometry of the target protein's binding pocket. Under the hood, the pipeline represents the docked protein-ligand complex as a sparse heterogeneous graph. It uses grouped vector attention to capture the spatial interactions between atoms and then passes these embeddings into a VAE-Transformer architecture to decode the new fragments as SMILES strings. The authors evaluate the method on the CrossDock2020 benchmark, reporting improvements over two baseline models in metrics like Vina binding scores, diversity, and chemical validity.

**Audience:**

Yes

**Audience Explanation:**

Despite the methology being incremental, I think there should always be some interesting towards the paper. I find the ablation experiments to be useful.

**Claims And Evidence:**

No

**Claims Explanation:**

- Lack of comprehensive evaluation: For a paper relying on a combination of existing methods, the empirical validation is too narrow. The experiments are restricted to a single dataset (CrossDock2020) and only compare against two main baselines (DiffSBDD and CFOM).
- Weak validation metrics: Relying almost entirely on classical Vina scores to claim state-of-the-art binding affinity is a rough approximation; the claims would be much stronger with a broader set of recent generative baselines or more rigorous physics-based scoring.

**Requested Changes:**

- Expand baseline comparisons: Comparing against only two models (DiffSBDD and CFOM) is insufficient to support claims of state-of-the-art performance. The authors must evaluate MODOLO against a wider array of recent 3D structure-based generative models or traditional fragment-growing algorithms.
- Broaden the evaluation datasets: Relying exclusively on a single split of the CrossDock2020 dataset  does not adequately demonstrate robustness or generalization. Evaluate the model on additional, distinct benchmarks (e.g., DUD-E, LIT-PCBA) or a set of diverse, manually curated challenging targets.
- Implement more rigorous affinity metrics: Autodock Vina scores are notoriously rough approximations of binding affinity and are insufficient as the sole proof of successful "optimization". The authors need to validate top generated candidates using more accurate methods, such as deep learning-based scoring functions (e.g., GNINA) or physics-based simulations (e.g., molecular dynamics or free energy perturbation).

---

> ### Author Response · Authors · 2026-03-30
>
> 1. Expanded Baseline Comparisons:
>
> We agree that comparing our method solely against DiffSBDD and CFOM was too narrow. To provide a more comprehensive comparison against recent generative models, we have now evaluated MODOLO against 2 additional baseline models, DiffDec and problem adjusted Pocket2Mol.
>
> 2. Broadened Evaluation Datasets:
>
> To demonstrate robustness and generalization beyond the single split of the CrossDock2020 dataset, we curated a new, challenging subset from the DUD-E dataset. This new evaluation set consists of 8 diverse proteins from different groups, each paired with 12 ligands, ensuring a more rigorous test across distinct targets. The results are in the appendix
>
>
> 3. Implemented More Rigorous Affinity Metrics:
>
> We have rerun all evaluations using GNINA. We now report the deep learning-based pose CNN score alongside our original metrics. The results of our method remain still stronger and statistically significant

---

### Review · Reviewer_uLWv · 2026-03-17

**Summary Of Contributions:**

The paper provides a novel technique for lead optimization, the process of adding small changes to a molecular scaffold in order to improve drug properties. A molecule is decomposed into a core scaffold and auxiliary fragments and a sophisticated conditional generation model samples new fragments. The conditioning takes both the scaffold and the protein pocket into consideration. The method is benchmarked against a couple of recent ML techniques on a standard evaluation setup for this problem.

**Additional Comments:**

Recommendations for improving performance:

Are ESM2 embeddings actually necessary? They're very high-dimensional. You may consider using much more basic physico-chemical descriptors or even one-hot encodings.

You resample fragments from the prior (conditioned on the pocket + scaffold). You should consider sampling that takes the input molecule candidate into consideration. Draw latent variable Z from P(Z | input candidate) and then sample from P(new fragment | Z), instead of sampling Z from the prior. I've done this sort of thing successfully in prior projects.

**Audience:**

Yes

**Audience Explanation:**

Lead optimization is a fundamental step in drug development, and the field is very keen to advance current virtual screening methods. The proposed technique is well motivated and interesting.

**Broader Impact Concerns:**

No concerns, beyond standard dual-use considerations for improving biotechnology methods. I don't think the paper should be expected to address these explicitly.

**Claims And Evidence:**

No

**Claims Explanation:**

The superiority of this method to the baselines is not clear. The error bars are very wide in the reported results. The results may not be statistically significant. Given that the evaluation set contains multiple protein targets and for each multiple samples are generated, you could do a paired statistical test, where you pair based on the target.

**Requested Changes:**

See above comments about the benchmarking and significance tests.

I found the paper's framing that it is doing 'optimization' a bit imprecise. For example, this sentence: "In contrast to these prediction-focused methods, our work addresses a generative task: learning to modify an existing molecule to better fit a given protein binding site-thereby optimizing for biological activity rather than just predicting it." If I understand correctly, your technique samples new fragments from a model that was not trained on data with any explicit notion of 'better'. It re-samples new fragments from a sophisticated conditional distribution and we hope that some of those samples are better. I'd either make the manuscript's framing more precise or change the training objective to use supervision about what makes something 'better'.

I found this claim surprising: "To prevent data leakage and ensure structural diversity, the dataset was clustered at 30% sequence identity using MMseqs2." This only considers diversity in protein space. Don't you also need diversity in molecule space?

I found the 'No VAE' ablation confusing. In what sense is it deterministic? What is the loss function? What is the architecture? Isn't it still a transformer that you can sample from (categorical value at each position)?
Recommendations for making results better
Are ESM2 embeddings actually necessary? They're very high-dimensional. You may consider using much more basic physico-chemical descriptors or even one-hot encodings.

You resample fragments from the prior (conditioned on the pocket + scaffold). You should consider sampling that takes the input molecule candidate into consideration. Draw latent variable Z from P(Z | input candidate) and then sample from P(new fragment | Z), instead of sampling Z from the prior. I've done this sort of thing successfully in prior projects.

---

> ### Author Response · Authors · 2026-03-30
>
> 1. Statistical Significance and Evaluation Benchmarking
>
> We conducted a paired statistical test on the Vina scores, and we show the results in the results tables in the paper.
>
> 2. Framing of "Optimization" vs. Generative Sampling
>
> In our work, the task is conditioned on a given reference molecule, and the goal is to produce new molecules that improve specific properties while maintaining similarity to the original structure. This aligns with the standard definition of molecule optimization, where the objective is to explore the local chemical space around a starting point under property constraints.
>
> 3. Dataset Leakage and Molecular Diversity
>
> The data split is based on paper (A 3D generative model for structure-based drug design) and is considered a standard when using crossdock2020. We have clarified it in section 4.1
>
> 4. Clarification on the 'No VAE' Ablation
>
> By "deterministic," we meant that the architecture forces the decoder to map the 3D structural context of the hole directly to a single SMILES sequence, effectively removing the continuous latent variable bottleneck. We fixed the sentence in section 4.4
>
> 5. ESM2 Embeddings Ablation
>
> An ablation was added to the results section
>
>
> 6. Alternative Latent Variable Sampling
>
> An ablation was added to the results section.

---

### Review · Reviewer_z1nL · 2026-03-25

**Summary Of Contributions:**

## Summary

The authors consider the scaffold completion task using a VAE framework on the CrossDocked dataset.

## Weaknesses

Overall, the paper lacks novelty, practically important results, and is fraught with serious methodological issues.

The authors state that they address the lead optimization problem and propose a novel framework for it. However, this problem has already been addressed in previous works ([1, 2, 14]), and there is no novelty or thoughtful modification of previous approaches. Moreover, the training objectives of the MOLODO model, the chosen metrics, the set of baselines, and their training objectives are not aligned with the lead optimization task.

### Context
Several pocket-conditioned molecular generative problems exist:
- (a) mimicking a pocket-conditional molecule distribution ([3, 12]);
- (b) pocket-conditional molecular property optimization ([4, 5, 6]);
- (c) lead optimization [1, 2].

### Issues:
1) **Misaligned training objective.** The actual training objective of MOLODO is more aligned with setting (a), with the difference being that the proposed setting is additionally conditioned on the scaffold. The authors approximately learn $p(\text{ligand} \mid \text{scaffold}(\text{ligand}), \text{pocket})$ (by minimizing the ELBO in a VAE framework), while the lead optimization problem is formulated as learning $p(\text{lead} \mid \text{reference}, \text{pocket})$ or $\max(\text{Affinity}(\text{lead} \mid \text{pocket}))$ s.t. $[\text{reference}, \text{pocket}, \text{distance}(\text{lead}, \text{reference}) < \text{threshold}]$. The training objectives of the baselines are also not related to lead optimization; for example, DiffSBDD was trained in setting (a).

2) **Misaligned metrics.** Problem settings (a), (b), and (c) differ in terms of the metrics used (see [3, 6]). The metrics provided by the authors are used for problem (b), yet the authors train a generative model in setting (a) (with the differences described above) while aiming to solve problem (c). To highlight, the used metrics reflect neither the properties of the lead in terms of improved affinity compared to the reference, nor the similarity to the reference.

3) **Outdated and non-representative baselines.** The authors use a non-representative and outdated set of baselines, ignoring most of the relevant ones. For example, the authors ignored [4, 5, 6, 10] (setting (b)), [9, 12, 13] (setting (a)), and [3, 7, 8, 11] (settings (a) and (b)). All these models can be straightforwardly adapted to problem (c) [2]. Notably, most of the mentioned baselines outperform MOLODO on the chosen metrics even in a zero-shot scenario.


[1] Simon Steshin. Lo-hi: Practical ml drug discovery benchmark. Advances in
Neural Information Processing Systems, 36:64526–64554, 2023.

[2] Duanhua Cao, Zhehuan Fan, Jie Yu, Mingan Chen, Xinyu Jiang, Xia Sheng,
Xingyou Wang, Xiaomin Luo, Dan Teng, and Mingyue Zheng. Benchmark-
ing real-world applicability of molecular generative models from de novo
design to lead optimization with molgenbench. bioRxiv, 2025.

[3] Arne Schneuing, Ilia Igashov, Adrian W Dobbelstein, Thomas Castiglione,
Michael M Bronstein, and Bruno Correia. Multi-domain distribution learn-
ing for de novo drug design. In The Thirteenth International Conference on
Learning Representations.

[4] Seonghwan Seo, Minsu Kim, Tony Shen, Martin Ester, Jinkyoo Park, Sung-
soo Ahn, and Woo Youn Kim. Generative flows on synthetic pathway for
drug design. In The Thirteenth International Conference on Learning Rep-
resentations, 2025.

[5] Tony Shen, Seonghwan Seo, Ross Irwin, Kieran Didi, Simon Olsson,
Woo Youn Kim, and Martin Ester. Compositional flows for 3d molecule
and synthesis pathway co-design. In Proceedings of the 42nd International
Conference on Machine Learning (ICML), 2025.

[6] Tony Shen, Seonghwan Seo, Grayson Lee, Mohit Pandey, Jason R Smith,
Artem Cherkasov, Woo Youn Kim, and Martin Ester. TacoGFN: Target-
conditioned GFlownet for structure-based drug design. Transactions on
Machine Learning Research, 2024.

[7] Xiwei Cheng, Xiangxin Zhou, Yuwei Yang, Yu Bao, and Quanquan Gu.
Decomposed direct preference optimization for structure-based drug design,
2025.

[8] Siyi Gu, Minkai Xu, Alexander Powers, Weili Nie, Tomas Geffner, Karsten
Kreis, Jure Leskovec, Arash Vahdat, and Stefano Ermon. Aligning target-
aware molecule diffusion models with exact energy optimization. Advances
in Neural Information Processing Systems, 37:44040–44063, 2024.

[9] Jiaqi Guan, Xiangxin Zhou, Yuwei Yang, Yu Bao, Jian Peng, Jianzhu Ma,
Qiang Liu, Liang Wang, and Quanquan Gu. Decompdiff: Diffusion models
with decomposed priors for structure-based drug design. In International
Conference on Machine Learning, pages 11827–11846. PMLR, 2023.

[10] Danny Reidenbach. EvoSBDD: Latent evolution for accurate and efficient
structure-based drug design. ICLR 2024 Workshop on Machine Learning
for Genomics Explorations, 2024.

[11] Xiangxin Zhou, Xiwei Cheng, Yuwei Yang, Yu Bao, Liang Wang, and Quan-
quan Gu. Decompopt: Controllable and decomposed diffusion models for
structure-based molecular optimization. In The Twelfth International Con-
ference on Learning Representations

[12] Jiaqi Guan, Wesley Wei Qian, Xingang Peng, Yufeng Su, Jian Peng, and
Jianzhu Ma. 3d equivariant diffusion for target-aware molecule generation
and affinity prediction. In International Conference on Learning Represen-
tations, 2023.

[13] Xingang Peng, Shitong Luo, Jiaqi Guan, Qi Xie, Jian Peng, and Jianzhu Ma.
Pocket2mol: Efficient molecular sampling based on 3d protein pockets. In
International conference on machine learning, pages 17644–17655. PMLR,
2022

[14] Soojung Yang, Doyeong Hwang, Seul Lee, Seongok Ryu, and Sung Ju
Hwang. Hit and lead discovery with explorative rl and fragment-based
molecule generation. Advances in Neural Information Processing Systems,
34:7924–7936, 2021.

**Audience:**

No

**Audience Explanation:**

The problem considered by the authors was already addressed in prior work. The authors offer no novelty or meaningful modifications to existing approaches, and the work is further hindered by significant methodological flaws.

**Broader Impact Concerns:**

-

**Claims And Evidence:**

No

**Claims Explanation:**

-

**Requested Changes:**

The paper requires a complete rework.

---

> ### Author Response · Authors · 2026-03-30
>
> We do not claim novelty for the problem of lead optimization itself, but rather introduce a novel architectural solution for this highly constrained setting. Unlike unconstrained de novo generation,  the problem we are tackling is generating a new molecule with improved properties such as affinity while remaining similar to the original molecule (lead optimization problem, which is also tackled by[1][2] or [3][4][5] which try to optimize different properties). This requires a fundamentally different architectural and evaluative approach, which we address below.
>
> Training Objective & Task Alignment
>
> While MOLODO optimizes a scaffold-conditioned objective, p(lead∣reference,pocket), rather than a full reference-conditioned one, this is standard practice in recent state-of-the-art lead optimization. For instance, CFOM [1] and DiffDec [2] explicitly decomposes leads into preserved cores and editable chains, while DiffSBDD utilizes partial diffusion to remain close to the origin. We will revise the introduction to clarify this formulation and clarify the baselines definitions to explain what configuration of DiffSBDD we use. It is very common in the state of the art today, to train in a similar scaffold conditioning method as we applied in our work.
>
> Metrics
>
> We respectfully disagree that our metrics do not reflect lead-optimization criteria. We explicitly measure both structural similarity to the reference (Tanimoto similarity via Morgan fingerprints) and affinity improvement. Our primary Success metric is strictly defined as:
> (Sim(ref,gen)>0.4)∧(Vina(gen)<Vina(ref)).
> We agree that our training objective is not the strict classical lead-optimization objective, but we would like to clarify that the evaluation is designed to capture key lead-optimization: preserving similarity to the starting ligand while improving predicted binding. We will revise the manuscript to make this distinction more explicit and avoid conflating the training formulation with the evaluation setting.
>
> Baselines and Task Alignment
>
> We thank the reviewer for suggesting additional literature and have expanded our empirical comparison to include DiffDec [2], a closely related scaffold-decoration baseline. Under a common evaluation protocol, DiffDec is competitive with MOLODO, whereas DiffSBDD performs substantially worse on the reference-relative metrics used here
> We caution against zero-shot cross-paper comparisons with the other suggested methods. Because they employ different generation settings, conditioning signals, and evaluation criteria, their reported metrics are not directly comparable to our joint success metric. To avoid misleading comparisons, we restrict quantitative claims to methods run under the exact same protocol. We will expand the discussion of these baselines and clarify these scope limitations in the manuscript.
>
> [1] CFOM: lead optimization for drug discovery with limited data. Kaminsky, Natan and Singer, Uriel and Radinsky, Kira.
>
> [2] DiffDec: structure-aware scaffold decoration with an end-to-end diffusion model.Xie, Junjie and Chen, Sheng and Lei, Jinping and Yang, Yuedong.
>
> [3]Unpaired Generative Molecule-to-Molecule Translation for Lead Optimization. Barshatski, Guy and Radinsky, Kira.
>
> [4] Generating Optimized Molecules without Patent Infringement.Turutov, Sally and Radinsky, Kira.
>
> [5] Hierarchical Generation of Molecular Graphs using Structural Motifs.Jin, Wengong and Barzilay, Dr.Regina and Jaakkola, Tommi.

---

> > ### Comment · Reviewer_z1nL · 2026-04-01
> >
> > "We do not claim novelty for the problem of lead optimization itself, but rather introduce a novel architectural solution for this highly constrained setting."
> > 1.1) In the contributions section, you state: "We propose MODOLO, a novel 3D-aware generative framework for lead optimization ...".
> >       In my earlier summary, I noted: "The authors state that they address the lead optimization problem and propose a novel framework for it".
> >       I consider that there is no novelty in your framework, nor any thoughtful modification of previous approaches [1, 2, 14].
> > 1.2) Given that your primary contribution is claimed to be architectural, several points remain unclear:
> >   1.2.1) What precisely constitutes the conceptual architectural novelty compared to the architectures in [1-14]? Please provide a detailed comparison with each.
> >   1.2.2) Why is a review of existing architectural solutions absent from the paper?
> >   1.2.3) Why you not compare a performace of different architectures on this problem? From your response, I understand that this should be the main experiment.
> > 1.3) You do not address lead optimization; you address the scaffold completion problem.
> >
> > Training Objective & Task Alignment
> > 2) In the paper (sec. 3.2, sec 4.1), you maximize p(ligand| scaffold(ligand, pocket)) instead of p(lead∣reference,pocket). Scaffold completion and lead optimization are distinct tasks.
> >
> > Metrics
> > 3) I noticed a success metric in the experimental results. Still, it is inadequate to consider only one problem-related metric with an arguable ligand similarity distance.
> > 3.1) Separate distance and improvement metrics should be added.
> > 3.2) What is the motivation behind the chosen threshold value? The commonly used value for Morgan fingerprints is around 0.8; an ablation study across different similarity levels is therefore necessary.
> > 3.3) The choice of Tanimoto distance is not justified. Distances in the latent space (e.g., Uni-Mol, ChemNet) should also be considered.
> >
> > Baselines and Task Alignment
> > 4) Due to misalignments between the problem statement, training objective, and evaluation metrics, it remains unclear which problem the article actually addresses: lead optimization or pocket-conditional scaffold completion.
> > In either case, the baselines are not representative. Current SOTA models for pocket-conditional generative problems (a) and (b) should be adapted and included.
> > Where necessary, they should be fine-tuned for the fixed-scaffold scenario, for instance via proper prompting for autoregressive models or partial diffusion in the case of diffusion-based models.
> > Additionally, a baseline that attaches fragments to the scaffold at random should be added.

---

> > > ### Author Response · Authors · 2026-04-18
> > > **reply to 1**
> > >
> > > Papers [1] and [2] are primarily benchmark papers that define evaluation settings; they do not propose our modeling approach. Reference [14] is methodologically closer, but still substantially different: it uses iterative RL-based fragment growing, lacks inference-time pocket conditioning, and requires target-specific training. In contrast, *MODOLO* jointly conditions on both the lead-derived scaffold and the protein pocket, performs *one-shot generation*, preserves the core explicitly, and generalizes to unseen targets *without retraining*. These are substantive modeling differences, not merely implementation details. We therefore kindly disagree that there is no architectural novelty. Nevertheless, we agree the term framework might be considered too broad and we revised the paper novelty claim to be more precise. In particular, we made it clear that our contribution is *not* the introduction of lead optimization as a task, nor the definition of a new benchmark. Rather, our contribution is a *distinct 3D-aware, protein-conditioned generative model for lead optimization*, with design choices that differ materially from prior work.
> > >
> > >
> > > We have extended our related work section to compare to the architectures and added the relevant ones (Pocket2Mol) to our baselines. Below is a full description of the architectures and how they relate to our problem.
> > >
> > >
> > > 1. Evaluation Frameworks vs. Generative Architectures Works such as Lo-hi [1] and MolGenBench [2] are not generative models but diagnostic ecosystems designed to evaluate Hit-to-Lead (H2L) capabilities and strict conformational validity. MODOLO provides the direct algorithmic solution to the paradigms these benchmarks define. By algorithmically preserving the active scaffold and generating mathematically valid SMILES sequences, MODOLO natively satisfies the rigid structural requirements that these frameworks test for.
> > > 2. 3D Coordinate Diffusion and Flow Matching vs. 1D Sequence Decoding A major divergence exists between MODOLO and models relying on continuous 3D coordinate generation, such as TargetDiff [12], DrugFlow [3], DecompDiff [9], DecompDiff [11], CGFlow [5], AliDiff [8] and DecompDPO [7]. These architectures iteratively denoise Gaussian noise or sample from prior distributions to generate point clouds or 3D graphs representing the ligand.
> > > The Difference: Continuous 3D generation is computationally heavy and often requires post-hoc heuristics to determine chemical validity. MODOLO elegantly avoids this by extracting the reference scaffold and encoding the 3D target environment via a Graph Neural Network (GNN), but then maps this spatial context to a 1D domain. By decoding the new fragment via a Transformer as a SMILES sequence in a single pass, MODOLO completely circumvents the iterative denoising latency and guarantees topological validity.
> > > 3. Closed-Set Fragment Assembly vs. Open Latent Space Generation Several Reinforcement Learning (RL) architectures restrict their generative space to predefined components. RxnFlow [4], TacoGFN [6], and FREED [14] assemble molecules iteratively from a closed set of known fragments or synthetic pathway steps.
> > > The Difference: This rigid library constraint limits the models' ability to invent out-of-distribution structural motifs. Additionally, models like RxnFlow [4] and FREED [14] lack target awareness during the generative inference step, requiring external training or post-hoc filtering (e.g., docking thousands of samples [4]). MODOLO operates on a continuous latent space and decodes fragments openly via its Transformer. Because its GNN directly incorporates the 3D pocket embedding during inference, MODOLO acts as a highly target-aware, zero-shot generator capable of producing unbounded, customized fragments.
> > > 4. External Oracle Search vs. Internal Target Conditioning Models like EVOSBDD [10] and FREED [14] are not aware of the target protein and either require extensive use of external oracles [10] or a specific training for each target [14].
> > > 5. 3D Atom-by-Atom Autoregression vs. Fragment-Level Decoding Pocket2Mol [13] operates iteratively in 3D space, predicting atomic coordinates one atom at a time.
> > > The Difference: Spatial autoregression is highly susceptible to compounding geometric errors as the molecule grows. MODOLO replaces 3D atom-by-atom prediction with a 1D fragment-level sequence prediction. The Transformer outputs the entire required topology as a string, immune to the cascading Euclidean spatial drift that limits Pocket2Mol.

---

> ### Author Response · Authors · 2026-04-18
> **reply for 2**
>
> We agree that our original phrasing was too broad, and we revised it for greater precision. In particular, we made it clear that our contribution is *not* the introduction of lead optimization as a task, nor the definition of a new benchmark. Rather, our contribution is a *distinct 3D-aware, protein-conditioned generative model for lead optimization*, with design choices that differ materially from prior work.
>
>
> At the same time, we respectfully disagree with the implication that our formulation falls outside lead optimization altogether. In the recent literature, lead optimization is commonly treated as a broad family of structure-modification problems, within which scaffold-, fragment-, linker-, and sidechain-conditioned generation are all standard subtypes rather than unrelated tasks. For example, recent work explicitly frames lead optimization as conditional molecular generation under structural constraints, and includes scaffold-based optimization and scaffold hopping within that setting [a–c]. Thus, while our formulation is more specific than the most general objective (p(lead|reference, pocket)), it remains a standard *core-preserving subtype of lead optimization*, rather than a fundamentally different task.
> More specifically, our method conditions on a scaffold extracted from the input ligand, together with the target pocket, in order to preserve the privileged core while optimizing the remaining structure. This defines a constrained lead-optimization setting, not an unconstrained one. We agree that this distinction should be stated more clearly in the manuscript, and we will revise the text accordingly.
> To make this precise in the revision, we made the following changes:
> 1. **Abstract and Introduction:** We revised the wording to remove any implication that we claim novelty at the level of the overall lead-optimization problem or benchmark setting.
> 2. **Related Work:** We strengthened the comparison to prior work, explicitly distinguishing our approach from benchmark-oriented papers [1,2] and from additional approaches such as [14].
> 3. **Contribution statement:** we now state explicitly that the paper proposes a **new model design within an established lead-optimization setting**, rather than a new task definition.
> Accordingly, we agree with the reviewer that the manuscript needed to better calibrate the scope of its novelty claim, and we have revised it to do so. However, we respectfully disagree with the stronger conclusion that the method lacks novelty altogether. The novelty of **MODOLO** lies not in redefining the task, but in the proposed **protein-conditioned, 3D-aware model design** for this established and practically important constrained lead-optimization setting.
>
>
> [a]-Deep lead optimization: leveraging generative AI for structural modification, Odin Zhang, Journal of the American Chemical Society
>
> [b]- Lin et al., CBGBench: Fill in the Blank of Protein-Molecule Complex Binding Graph (2024) — scaffold/fragment/linker/sidechain tasks in a unified drug-design benchmark.
>
> [c]- Huang, L., Xu, T., Yu, Y. et al. A dual diffusion model enables 3D molecule generation and lead optimization based on target pockets. Nat Commun 15, 2657 (2024).

---

> ### Author Response · Authors · 2026-04-18
> **reply for 3-4**
>
> 3)
>
> We thank the reviewer for this suggestion and have revised the evaluation accordingly.
>
> First, we report not only the aggregate success metric, but also its two underlying components separately: binding affinity improvement and ligand similarity. These quantities are now included throughout the main tables and ablation studies, so the reader can directly assess the tradeoff between optimization quality and structural preservation rather than relying on a single composite metric.
>
> Second, we added an ablation over multiple similarity thresholds. This analysis is included for both Morgan fingerprint similarity and ChemNet similarity, and directly addresses the reviewer’s concern regarding the sensitivity of the results to the threshold choice.
>
> Third, following the reviewer’s suggestion, we added an additional ablation using ChemNet-based similarity (Section D), alongside the original Morgan/Tanimoto-based evaluation. This allows us to evaluate our method under both a standard fingerprint-based similarity measure and a learned latent-space similarity measure.
>
> Finally, we expanded the discussion motivating the threshold values used in our experiments and added citations to prior work using related similarity ranges (section 4.5). We also added an ablation over different scaffold constructions, further strengthening the analysis of how structural constraints affect performance.
>
> 4.
>
> For (a), DiffSBDD is already a direct baseline and, in fact, corresponds to the “partial diffusion” approach suggested by the reviewer. In its original formulation, DiffSBDD performs optimization by partially noising and denoising a reference molecule, followed by selection under the desired objective. We used exactly this optimization procedure in our original submission. We now make this correspondence explicit in the main text to avoid ambiguity.
>
> To cover the autoregressive family of pocket-conditioned generators requested by the reviewer, we additionally included Pocket2Mol in the revised manuscript. Pocket2Mol generates molecules autoregressively, atom-by-atom, conditioned on the binding pocket. To adapt it to our fixed-scaffold setting, we fine-tuned it accordingly; the protocol is described in Appendix B.
>
> For (b), we added PMDM as an additional strong baseline. PMDM is a recent pocket-conditioned diffusion model designed for target-aware 3D ligand generation, and its original paper further demonstrates its use in lead optimization, including scaffold-/fragment-constrained generation settings. We therefore view PMDM as an appropriate representative baseline for the scaffold-constrained completion side of the reviewer’s request. For fairness, PMDM is given the exact same extracted scaffold used by our method.
>
> Finally, as requested, we added a random attachment baseline. Specifically, we attach randomly sampled sidechains/fragments from the empirical training distribution to the scaffold.

---

> ### Comment · Reviewer_z1nL · 2026-04-29
> **Part 1**
>
> In the paper, the authors, as they stated, propose a novel model for a particular molecular generative problem - lead optimization and achieve SOTA quality on this task.
> In my opinion, none of these statements are valid:
>
> 1) The model architecture is fairly common and lacks novelty.
> 2) The problem that the authors actually addressed is scaffold completion.
> 3) The statements about superior quality are vague because of inadequate and provably erroneous comparisons with related methods.
>
> In detail:
>
> 1) The proposed MOLODO model consists of a geometric attention-based pocket encoder and an MLN (molecular line notation) transformer ligand decoder.
> There are plenty of works that use geometric attention-based pocket encoders [3, 4, 6, 12, 13, 17, 18, 19, 21].
> The MLN Transformer decoder is also extensively used in molecular generation [16, 17, 18, 19].
> The conditioning scheme is also not novel and was previously used in [6, 13, 14, 15, 20, 21].
>
> 2) Throughout the paper, the authors misunderstood scaffold completion as the lead optimization task.
> Note that lead optimization can be framed as scaffold completion if training of the model is performed on an appropriate dataset, which consists of pairs of leads and their scaffolds.
> The authors do not construct such a dataset nor give an example of one that already exists.
> Instead, the authors provide experiments on the CrossDocked dataset, which is not related to lead optimization, while stating throughout the paper that their model solves lead optimization.
>
> 3) Issues:
>     1) If the authors claim that their contribution is architectural, in the main experiment they should compare different architectures on a specific setting.
> This means that all aspects such as training framework (VAE, autoregression, diffusion), optimizer, etc., should be fixed except for the actual model architecture.
> Such an experiment is absent in the paper; instead, the authors compare overall generative frameworks on the scaffold completion task.
> Moreover, a proper analysis of architectures of molecular generative models is absent in the paper.
>     2) Baseline models should be properly adapted for scaffold completion, i.e., they should be retrained with a proper conditioning scheme.
> It is not honest to compare models trained to generate molecules de novo (even with proper prompt, e.g., in a zero-shot scenario) on the scaffold completion task with a model trained to complete scaffolds.
> For example, the DiffSBDD model is used as is, without retraining.
> The authors state that they retrain the Pocket2Mol model; however, I have doubts that this was done properly, based on the reported metrics.
> Since de novo generation is much harder than the scaffold completion task, one could expect better distribution reconstruction quality (i.e., difference of the reported metrics with the reference set); however, the results are the opposite.
> Moreover, the dispersion of the reported docking-based metrics is huge (which is not observed in previous works), indicating errors in the evaluation protocol.
>     3) The baselines are not representative.
> Most of the SOTA pocket-conditional molecular generative models (de novo pocket-conditional probabilistic or optimization-based models) are absent from the comparison and ignored in the related work review.
> As mentioned previously, adaptation for most of the models is fairly straightforward.
>     4) Since the authors train their MOLODO model to reconstruct a molecule from its scaffold, the main metrics should reflect the model's ability to recover the conditional distribution.
> FCD, various divergences between distributions of reference and generated molecules for descriptors such as QED, SA, RB, logP, etc., should be taken into account (for more details, see [3]).
> I want to highlight that there is not much sense in comparing models trained to recover distributions based on absolute values of affinity or drug-likeness of generated molecules.
> Instead, one should compute how different these values are compared to the reference set of molecules.
>     5) The diversity metric is ignored.
> This metric is crucial in any generative task and commonly conflicts with fidelity or reconstruction quality, similar to the precision-recall tradeoff.
> MOLODO performs quite poorly in terms of diversity, and ignorance of this metric also contradicts the statement about superiority of the proposed model.
> Interestingly, diversity was reported in the first revision and later disappeared entirely from the paper.
>     6) If the authors want to address the lead optimization problem, they should consider optimization-based baseline methods and use corresponding metrics.
> [4, 5, 6, 7, 8] baselines should be added.
>     7) Validity and novelty metrics should also be taken into account.

---

> > ### Comment · Reviewer_z1nL · 2026-04-29
> > **Part 2**
> >
> > 3.8) As mentioned previously, the reported values of standard deviation for the docking-related metrics are huge.
> > These results contradict values reported in previous works, indicating errors in the evaluation protocol.
> > This observation makes me skeptical overall about the reported numbers in the paper and their further interpretation.
> >
> > To summarize
> > All my concerns about the paper are still present. I believe the paper lacks any insights or practically important results.
> >
> >
> > [1] Simon Steshin. Lo-hi: Practical ml drug discovery benchmark. Advances in
> > Neural Information Processing Systems, 36:64526–64554, 2023.
> >
> > [2] Duanhua Cao, Zhehuan Fan, Jie Yu, Mingan Chen, Xinyu Jiang, Xia Sheng,
> > Xingyou Wang, Xiaomin Luo, Dan Teng, and Mingyue Zheng. Benchmark-
> > ing real-world applicability of molecular generative models from de novo
> > design to lead optimization with molgenbench. bioRxiv, 2025.
> >
> > [3] Arne Schneuing, Ilia Igashov, Adrian W Dobbelstein, Thomas Castiglione,
> > Michael M Bronstein, and Bruno Correia. Multi-domain distribution learn-
> > ing for de novo drug design. In The Thirteenth International Conference on
> > Learning Representations.
> >
> > [4] Seonghwan Seo, Minsu Kim, Tony Shen, Martin Ester, Jinkyoo Park, Sung-
> > soo Ahn, and Woo Youn Kim. Generative flows on synthetic pathway for
> > drug design. In The Thirteenth International Conference on Learning Rep-
> > resentations, 2025.
> >
> > [5] Tony Shen, Seonghwan Seo, Ross Irwin, Kieran Didi, Simon Olsson,
> > Woo Youn Kim, and Martin Ester. Compositional flows for 3d molecule
> > and synthesis pathway co-design. In Proceedings of the 42nd International
> > Conference on Machine Learning (ICML), 2025.
> >
> > [6] Tony Shen, Seonghwan Seo, Grayson Lee, Mohit Pandey, Jason R Smith,
> > Artem Cherkasov, Woo Youn Kim, and Martin Ester. TacoGFN: Target-
> > conditioned GFlownet for structure-based drug design. Transactions on
> > Machine Learning Research, 2024.
> >
> > [7] Xiwei Cheng, Xiangxin Zhou, Yuwei Yang, Yu Bao, and Quanquan Gu.
> > Decomposed direct preference optimization for structure-based drug design,
> > 2025.
> >
> > [8] Siyi Gu, Minkai Xu, Alexander Powers, Weili Nie, Tomas Geffner, Karsten
> > Kreis, Jure Leskovec, Arash Vahdat, and Stefano Ermon. Aligning target-
> > aware molecule diffusion models with exact energy optimization. Advances
> > in Neural Information Processing Systems, 37:44040–44063, 2024.
> >
> > [9] Jiaqi Guan, Xiangxin Zhou, Yuwei Yang, Yu Bao, Jian Peng, Jianzhu Ma,
> > Qiang Liu, Liang Wang, and Quanquan Gu. Decompdiff: Diffusion models
> > with decomposed priors for structure-based drug design. In International
> > Conference on Machine Learning, pages 11827–11846. PMLR, 2023.
> >
> > [10] Danny Reidenbach. EvoSBDD: Latent evolution for accurate and efficient
> > structure-based drug design. ICLR 2024 Workshop on Machine Learning
> > for Genomics Explorations, 2024.
> >
> > [11] Xiangxin Zhou, Xiwei Cheng, Yuwei Yang, Yu Bao, Liang Wang, and Quan-
> > quan Gu. Decompopt: Controllable and decomposed diffusion models for
> > structure-based molecular optimization. In The Twelfth International Con-
> > ference on Learning Representations
> >
> > [12] Jiaqi Guan, Wesley Wei Qian, Xingang Peng, Yufeng Su, Jian Peng, and
> > Jianzhu Ma. 3d equivariant diffusion for target-aware molecule generation
> > and affinity prediction. In International Conference on Learning Represen-
> > tations, 2023.
> >
> > [13] Xingang Peng, Shitong Luo, Jiaqi Guan, Qi Xie, Jian Peng, and Jianzhu Ma.
> > Pocket2mol: Efficient molecular sampling based on 3d protein pockets. In
> > International conference on machine learning, pages 17644–17655. PMLR,
> > 2022
> >
> > [14] Soojung Yang, Doyeong Hwang, Seul Lee, Seongok Ryu, and Sung Ju
> > Hwang. Hit and lead discovery with explorative rl and fragment-based
> > molecule generation. Advances in Neural Information Processing Systems,
> > 34:7924–7936, 2021.
> >
> > [15] Alexander Telepov, Artem Tsypin, Kuzma Khrabrov, Sergey Yakukhnov,
> > Pavel Strashnov, Petr Zhilyaev, Egor Rumiantsev, Daniel Ezhov, Manvel
> > Avetisian, Olga Popova, and Artur Kadurin. FREED++: Improving RL
> > agents for fragment-based molecule generation by thorough reproduction.
> > Transactions on Machine Learning Research, 2023.
> >
> > [16] Sanjar Adilov. Generative pre-training from molecules. 2021.
> >
> > [17] Ross Irwin, Spyridon Dimitriadis, Jiazhen He, and Esben Jannik Bjerrum.
> > Chemformer: a pre-trained transformer for computational chemistry. Ma-
> > chine Learning: Science and Technology, 3(1):015022, jan 2022.
> >
> > [18] Arne Schneuing, Charles Harris, Yuanqi Du, Kieran Didi, Arian Jamasb,
> > Ilia Igashov, Weitao Du, Carla Gomes, Tom L Blundell, Pietro Lio, et al.
> > Structure-based drug design with equivariant diffusion models. Nature Com-
> > putational Science, 4(12):899–909, 2024.

---

> > > ### Comment · Reviewer_z1nL · 2026-04-29
> > > **Part 3**
> > >
> > > [18] Jike Wang, Rui Qin, Mingyang Wang, Meijing Fang, Yangyang Zhang,
> > > Yuchen Zhu, Qun Su, Qiaolin Gou, Chao Shen, Odin Zhang, Zhenxing
> > > Wu, Dejun Jiang, Xujun Zhang, Huifeng Zhao, Jingxuan Ge, Zhourui Wu,
> > > Yu Kang, Chang-Yu Hsieh, and Tingjun Hou. Token-mol 1.0: tokenized
> > > drug design with large language models. Nature Communications, 16(1),
> > > May 2025.
> > >
> > > [19] Kehan Wu, Yingce Xia, Pan Deng, Renhe Liu, Yuan Zhang, Han Guo, Yu-
> > > meng Cui, Qizhi Pei, Lijun Wu, Shufang Xie, Si Chen, Xi Lu, Song Hu,
> > > Jinzhi Wu, Chi-Kin Chan, Shawn Chen, Liangliang Zhou, Nenghai Yu, En-
> > > hong Chen, Haiguang Liu, Jinjiang Guo, Tao Qin, and Tie-Yan Liu. Tam-
> > > gen: drug design with target-aware molecule generation through a chemical
> > > language model. Nature Communications, 15(1), October 2024.
> > >
> > > [20] Odin Zhang, Jintu Zhang, Jieyu Jin, Xujun Zhang, RenLing Hu, Chao Shen,
> > > Hanqun Cao, Hongyan Du, Yu Kang, Yafeng Deng, Furui Liu, Guangyong
> > > Chen, Chang-Yu Hsieh, and Tingjun Hou. Resgen is a pocket-aware 3d
> > > molecular generation model based on parallel multiscale modelling. Nature
> > > Machine Intelligence, 5(9):1020–1030, 2023.
> > >
> > > [21] Meng Liu, Youzhi Luo, Kanji Uchino, Koji Maruhashi, and Shuiwang Ji.
> > > Generating 3d molecules for target protein binding. In International Con-
> > > ference on Machine Learning, pages 13912–13924. PMLR, 2022.

---

### Decision · Action_Editor_cVef · 2026-05-09

**Recommendation:** Reject

**Audience:**

No

**Audience Explanation:**

The current findings are of limited interest to the TMLR audience due to substantive methodological flaws and a lack of architectural novelty.

Suggestions for the major revision:

1. For the work to be relevant to researchers in the field, the authors must undertake a complete rework that corrects the fundamental misunderstanding of lead optimization as scaffold completion. This requires rerunning experiments from the ground up to ensure a fair comparison by retraining baselines under a proper conditioning scheme.

2. Additionally, interest would depend on the inclusion of more representative baselines and a broader set of rigorous metrics.

**Claims And Evidence:**

No

**Claims Explanation:**

The claims in this submission are not supported by convincing evidence due to a fundamental misalignment between the stated task and the methodology. The authors conflate lead optimization with scaffold completion , utilizing a training objective that maximizes $p(\text{ligand} | \text{scaffold}(\text{ligand}, \text{pocket}))$ instead of the lead-relative distribution $p(\text{lead} | \text{reference}, \text{pocket})$. Furthermore, the empirical results are compromised by an unfair comparison: baseline models were evaluated in a zero-shot or poorly adapted manner rather than being retrained with the same conditioning scheme as the proposed model.

I would like to recommend a major revision for this paper, and consider involving an additional expert reviewer in the next round of review.

**Resubmission Of Major Revision:**

The authors may consider submitting a major revision at a later time.